# Population response magnitude variation in inferotemporal cortex predicts image memorability

Andrew Jaegle[1†], Vahid Mehrpour[1†], Yalda Mohsenzadeh[2,4,3†], Travis Meyer[1], Aude Oliva[2], Nicole Rust[1]*

[1]Department of Psychology, University of Pennsylvania, Philadelphia, United States; [2]Computer Science and Artificial Intelligence Laboratory, Massachusetts Institute of Technology, Cambridge, United States; [3]Department of Computer Science, Western University, London, Canada; [4]Brain and Mind Institute, Western University, London, Canada

**Abstract** Most accounts of image and object encoding in inferotemporal cortex (IT) focus on the distinct patterns of spikes that different images evoke across the IT population. By analyzing data collected from IT as monkeys performed a visual memory task, we demonstrate that variation in a complementary coding scheme, the magnitude of the population response, can largely account for how well images will be remembered. To investigate the origin of IT image memorability modulation, we probed convolutional neural network models trained to categorize objects. We found that, like the brain, different natural images evoked different magnitude responses from these networks, and in higher layers, larger magnitude responses were correlated with the images that humans and monkeys find most memorable. Together, these results suggest that variation in IT population response magnitude is a natural consequence of the optimizations required for visual processing, and that this variation has consequences for visual memory.
DOI: https://doi.org/10.7554/eLife.47596.001

*For correspondence:
nrust@psych.upenn.edu

[†]These authors contributed equally to this work

**Competing interests:** The authors declare that no competing interests exist.

## Introduction

At higher stages of visual processing such as inferotemporal cortex (IT), representations of image and object identity are thought to be encoded as distinct patterns of spikes across the IT population, consistent with neurons that are individually 'tuned' for distinct image and object properties. In a population representational space, these distinct spike patterns translate into population response vectors that point in different directions, and information about object identity is formatted such that it can be accessed from IT neural responses via a weighted linear decoder (*Figure 1a*; reviewed by *DiCarlo et al., 2012*). The magnitude of the IT population response is often assumed to be unimportant (but see *Chang and Tsao, 2017*), and it is typically disregarded in population-based approaches, including population decoding and representational similarity analyses (*Kriegeskorte et al., 2008*). Building on that understanding, investigations of cognitive processes, such as memory, appreciate the importance of equating image sets for the robustness of their underlying visual representations in an attempt to isolate the cognitive process under investigation from variation due to changes in the robustness of the sensory input. This process amounts to matching decoding performance or representational similarity between sets of images, in order to control for low-level factors (e.g. contrast, luminance and spatial frequency content) and visual discriminability (*Willenbockel et al., 2010*). Here we demonstrate that variation in IT population response magnitude has important behavioral consequences for one higher cognitive process: how

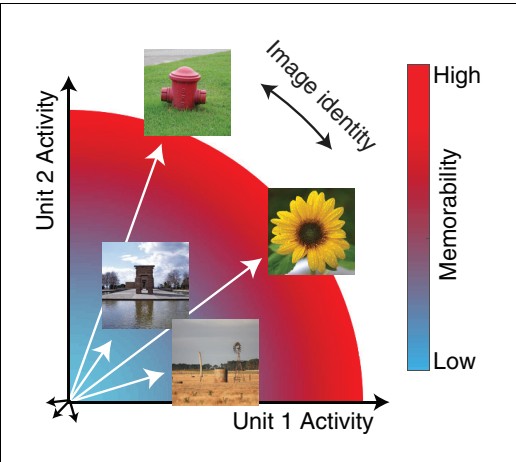

**Figure 1.** The hypothesis: the magnitude of the IT population response encodes image memorability. In geometric depictions of how IT represents image identity, the population response to an image is depicted as a vector in an N-dimensional space, where N indicates the number of neurons in the population, and identity is encoded by the direction of the population vector. Here we test the hypothesis that image memorability is encoded by the magnitude (or equivalently length) of the IT population vector, where images that produce larger population responses are more memorable.

DOI: https://doi.org/10.7554/eLife.47596.002

well images will be remembered. Our results suggest that the lack of appreciation for this type of variation in IT population response should be reconsidered.

'Image memorability' refers to the simple notion that some images are easy to remember while others are easy to forget (*Isola et al., 2011*). While a large component of image memorability variation is consistent across different individuals (*Isola et al., 2011*; *Khosla et al., 2015*), a full account of the sources of image memorability has remained elusive. The neural correlates of memorability are likely to reside at higher stages of the visual form processing pathway, where image memorability can be decoded from human fMRI activity patterns (*Bainbridge et al., 2017*; *Bainbridge and Rissman, 2018*), as well as with a consistent cortical timescale from human MEG data at ~150 msec (*Mohsenzadeh et al., 2019*). Linear decodability could imply that information about image memorability is represented in the same fashion as information about object identity: as population response vectors that point in different directions (*Figure 1a*; *DiCarlo et al., 2012*). However, under this proposal, it is not clear how our experience of image identity and image memorability would be represented by the same neural populations, for example the fact that one image of a person can be more memorable than another image of that same person. Here we present an alternative proposal, hinted at by the fact that more memorable images evoke larger fMRI responses (*Bainbridge et al., 2017*): we propose that memorability variation is determined principally by the magnitude of the IT population response (*Figure 1*). This scenario incorporates a representational scheme for memorability that is orthogonal to the scheme thought to support object identity, and if correct, would provide a straightforward account of how a high-level visual brain area such as inferotemporal cortex (IT) multiplexes visual information about image content (as the population vector direction) as well as memorability (as population vector magnitude). In an earlier report, we tested the hypothesis that changes in the lengths of IT population response vectors with stimulus repetition ('repetition suppression') could account for rates of remembering and forgetting as a function of time (*Meyer and Rust, 2018*). However, that work explicitly assumed that the population response vectors corresponding to different images were the same length (see Methods), whereas here we focus on whether variation in the lengths of these vectors can account for a previously undocumented behavioral signature in monkeys: image memorability.

## Results

To test the hypothesis presented in *Figure 1*, we obtained image memorability scores by passing images through a model designed to predict image memorability for humans (*Khosla et al., 2015*). The neural data, also reported in *Meyer and Rust (2018)*, were recorded from IT as two rhesus monkeys performed a single-exposure visual memory task in which they reported whether images were novel (never before seen) or were familiar (seen once previously; *Figure 2a*). In each experimental session, neural populations with an average size of 26 units were recorded, across 27 sessions in total. Because accurate estimate of population response magnitude requires many hundreds of units, data were concatenated across sessions into a larger pseudopopulation in a manner that aligned images with similar memorability scores (see Methods and *Figure 2—figure supplement 1*).

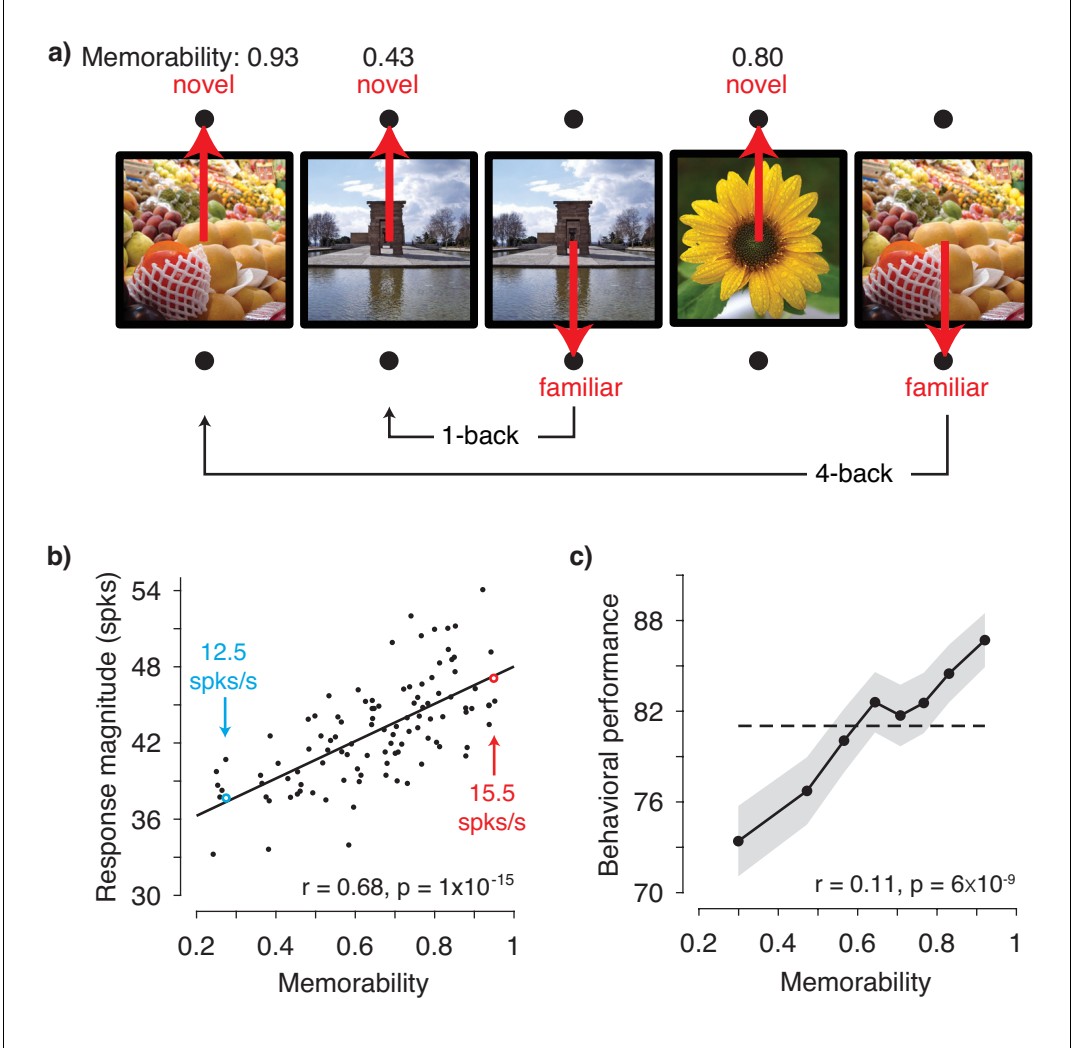

**Figure 2.** IT population response magnitude strongly correlates with image memorability. (a) The monkeys' task involved viewing each image for 400 ms and then reporting whether the image was novel or familiar with an eye movement to one of two response targets. The probability of a novel versus familiar image was fixed at 50% and images were repeated with delays ranging from 0 to 63 intervening trials (4.5 s to 4.8 min). Shown are 5 example trials with image memorability scores labeled. The memorability of each image was scored from 0-1, where the score reflects the predicted chance-corrected hit rate for detecting a familiar image (i.e., 0 maps to chance and 1 maps to ceiling; *Khosla et al., 2015*). (b) The relationship between image memorability scores and IT population response magnitudes. Each point corresponds to a different image (N=107 images). Population response magnitudes were computed as the L2 norm $\left(\sqrt{\sum_{i=1}^{N} r_i^2}\right)$, where $r_i$ is the spike count response of the ith unit, across a pseudopopulation of 707 units. Spikes were counted in an 80 ms window positioned 180 to 260 ms following stimulus onset (see *Figure 2—figure supplement 3a* for different window positions). The Pearson correlation and its p-value are labeled. The solid line depicts the linear regression fit to the data. For reference, the mean firing rates for two example images are also labeled (see also *Figure 2—figure supplement 3b*). (c) Mean and standard error (across experimental sessions) of monkey behavioral performance on the memory task as a function of human-based image memorability scores. For visualization, performance was binned across images with neighboring memorability scores and pooled across monkeys (see *Figure 2—figure supplement 4* for plots by individual). The dashed line corresponds to the grand average performance, and if there were no correlation, all points should fall near this line. The point-biserial correlation and its p-value, computed for the raw data (i.e. 2889 continuous memorability scores and 2889 binary performance values for each image in each session) are labeled. Source data are included as *Figure 2—source data 1* and *Figure 2—source data 2*.

DOI: https://doi.org/10.7554/eLife.47596.003

*Figure 2 continued on next page*

*Figure 2 continued*

The following source data and figure supplements are available for figure 2:

**Source data 1.** Data used to compute monkey neural responses as well as human-based memorability scores for each image.

DOI: https://doi.org/10.7554/eLife.47596.008

**Source data 2.** Data used to compute monkey behavioral responses as well as human-based memorability scores for each image.

DOI: https://doi.org/10.7554/eLife.47596.009

**Figure supplement 1.** Distributions of memorability scores for the images used in these experiments.

DOI: https://doi.org/10.7554/eLife.47596.004

**Figure supplement 2.** The correlation of memorability and population response magnitude, for each monkey individually.

DOI: https://doi.org/10.7554/eLife.47596.005

**Figure supplement 3.** The correlation of memorability and the IT population response, applied to different time windows, assessed with firing rate, and determined with top-ranked firing units removed.

DOI: https://doi.org/10.7554/eLife.47596.006

**Figure supplement 4.** Human-based memorability scores predict what monkeys find memorable.

DOI: https://doi.org/10.7554/eLife.47596.007

The resulting pseudopopulation contained the responses of 707 IT units to 107 images, averaged across novel and familiar presentations.

*Figure 2b* shows the correlation between image memorability and IT population response magnitudes, which was strong and highly significant (Pearson correlation: r = 0.68; p=$1\times10^{-15}$). This correlation remained strong when parsed by the data collected from each monkey individually (*Figure 2—figure supplement 2*) and, after accounting for the time required for signals to reach IT, across the entire 400 ms viewing period (*Figure 2—figure supplement 3a*). The correlation also remained strong when computed for a quantity closely related to response magnitude, grand mean firing rate (*Figure 2—figure supplement 3b*), as well as when the highest firing units were excluded from the analysis (*Figure 2—figure supplement 3c*). A strong correlation was also observed when images containing faces and/or bodies were excluded from the analysis (Pearson correlation: r = 0.62; p=$2\times10^{-10}$), suggesting that our results are not an artifactual consequence of recording from patches of neurons enriched for face or body selectivity (*Pinsk et al., 2005*; *Tsao et al., 2003*). Finally, at the same time that IT neural responses exhibited repetition suppression for familiar as compared to novel image presentations (mean proportional reduction in this spike count window = 6.2%; see also *Meyer and Rust, 2018*), the correlation remained strong when computed for the images both when they were novel (Pearson correlation: r = 0.62; p=$2\times10^{-12}$) as well as when they were familiar (Pearson correlation: r = 0.58; p=$8\times10^{-11}$).

The strength of the correlation between memorability and IT response magnitude is notable given the species difference, as the memorability scores were derived from a model designed to predict what humans find memorable whereas the neural data were collected from rhesus monkeys. In contrast to the human-based scores, which reflect the estimated average performance of ~80 human individuals, our monkey behavioral data are binary (i.e. correct/incorrect for each image). As such, the monkey behavioral data cannot be used in the same way to concatenate neural data across sessions to create a pseudopopulation sufficiently large to accurately estimate IT population response magnitudes. However, our data did allow us to evaluate whether human-based memorability scores were predictive of the images that the monkeys found most memorable during the single-exposure visual memory task, and we found that this was in fact the case (*Figure 2c*).

While the monkeys involved in these experiments were not explicitly trained to report object identity, they presumably acquired the ability to identify objects naturally over their lifetimes. The correlations between IT population response magnitude and image memorability could thus result from optimizations for visual memory, or it could follow more simply from the optimizations that support visual processing, including object and scene identification. If it were the case that a system trained to categorize objects and scenes (but not trained to report familiarity) could account for the correlations we observe between IT response magnitude variation and image memorability, this would suggest that image memorability follows from the optimizations for visual (as opposed to

mnemonic) processing. To investigate the origin of memorability variation, we investigated the correlate of memorability in a convolutional neural network (CNN) model trained to categorize thousands of objects and scenes but not explicitly trained to remember images or estimate memorability (*Khosla et al., 2015*). We found that the correlation between image memorability scores and their corresponding population response magnitudes was significantly higher in the trained as compared to a randomly initialized version of the network in all layers, and the strength of this correlation generally increased across the hierarchy (*Figure 3*). These results were also replicated in two other CNNs trained for object classification (*Krizhevsky et al., 2012*; *Simonyan and Zisserman, 2015*), where correlation strength also generally increased across the hierarchy of the network (*Figure 3—figure supplement 1*), suggesting that this signature is not unique to this particular architecture or training procedure. These results suggest that variation in population response magnitude across images is reflected in visual systems that are trained to classify objects, and that this variation is directly related to variation in image memorability.

## Discussion

Here we have demonstrated that variation in the ability of humans and monkeys to remember images is strongly correlated with the magnitude of the population response in IT cortex. These results indicate that memorability is reflected in IT via a representational scheme that lies largely orthogonal to the one IT has been presumed to use for encoding object identity (*Figure 1*). For example, investigations of how monkey IT and its human analogs represent objects using 'representational similarity analysis' typically begin by normalizing population response vector magnitude to be the same for all images such that all that is left is the direction of the population response pattern, under the assumption that population vector magnitude is irrelevant for encoding object or image identity (*Kriegeskorte et al., 2008*). Before our study, data from human fMRI had pinpointed the locus of memorability to the human analog of IT, but we did not understand 'how' the representations of memorable and non-memorable images differed. Our results point to a simple and coherent account of how IT multiplexes representations of visual and memorability information using two complementary representational schemes (*Figure 1*).

Investigations of cognitive processes such as memory have long appreciated the need to equate image sets for the robustness of their visual representations. The significance of our result follows from the unexpected finding that there is variation in the robustness of visual representations within the class of natural images that is not accounted for by classic population-based decoding approaches, and that this variation correlates with our understanding of the content that makes images more or less memorable. Our results demonstrate that despite the host of homeostatic mechanisms that contribute to maintaining constant global firing rates across a cortical population (*Turrigiano, 2012*), changes in image content can result in IT population response magnitudes that differ by ~20% (*Figure 2b*; *Figure 2—figure supplement 3b*). Future work will be required to explore the

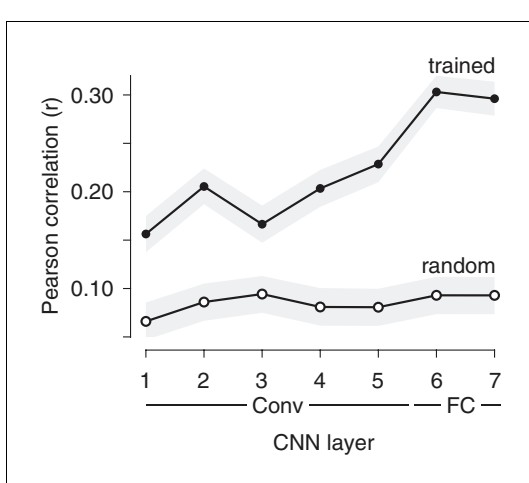

**Figure 3.** Correlations between memorability and population response increase in strength across layers of a CNN trained to classify objects and scenes. Shown are mean and 95% CIs of the Pearson correlations between image memorability and population response magnitude for each hierarchical layer of the CNN described in *Zhou et al. (2014)*, up to the last hidden layer. 'Conv': convolutional layer; 'FC': fully connected layer. p-values for a one-sided comparison that correlation strength was larger for the trained than the randomly connected network: p<0.0001 for all layers.
DOI: https://doi.org/10.7554/eLife.47596.010
The following figure supplement is available for figure 3:

**Figure supplement 1.** Correlations between memorability and population response magnitude are also reflected in two other CNNs.
DOI: https://doi.org/10.7554/eLife.47596.011

ultimate bounds of image memorability variation, possibly via the use of newly developed generative adversarial network models that create images with increased or decreased memorability (*Goetschalckx et al., 2019*).

Our work relates to 'subsequent memory effects' whereby indicators of higher neural activity in structures both within and outside the medial temporal lobe during memory encoding are predictive of more robust remembering later on (reviewed by *Paller and Wagner, 2002*). To tease apart whether the origin of memorability could be attributed to optimizations for visual as opposed to mnemonic processing, we investigated CNNs optimized to categorize objects but not explicitly trained to predict the memorability of images. While this class of models has been demonstrated to mimic many aspects of how IT represents visual object identity (reviewed by *Yamins and DiCarlo, 2016*), image memorability has a distinct representational scheme from identity (*Figure 1*). The fact that CNNs trained for object recognition mimic the neural representation of a distinct behavior – visual memorability – is compelling evidence that this strategy of multiplexing visual identity and memorability results from the computational requirements of optimizing for robust object representations. Our results are remarkably well-aligned with one study reporting the correlation between one CNN that we tested, VGG-16 (*Figure 3—figure supplement 1b*), and patterns of confusions during human rapid visual categorization behavior (*Eberhardt et al., 2016*). The correlation between what humans and monkeys find memorable (*Figure 2c*) is at first pass surprising in light of the presumed differences in what typical humans and monkeys experience. However, understanding that memorability variation emerges in CNNs trained for object categorization (*Figure 3*), coupled with the similarities in object representations between humans and monkeys (*Rajalingham et al., 2015*), provides insight into the preservation of memorability correlations across these two primate species.

The mechanism that we describe here is also likely to be partially but not entirely overlapping with descriptions of salience, where a relation between memorability and eye movement patterns exists (*Bylinskii et al., 2015*) and CNNs trained to categorize objects have been coupled with human fMRI responses to predict eye movement behavior (*O'Connell and Chun, 2018*). However, memorability effects are typically found even after controlling for factors commonly associated with salience, including images features and object categories (*Bainbridge et al., 2017*; *Bainbridge et al., 2013*; *Mohsenzadeh et al., 2019*), suggesting that memorability and salience are unlikely to be one and the same. For example, (*Bainbridge et al., 2017*; *Bainbridge et al., 2013*) reported differences in memorability between face stimuli that were identical in saliency in terms of their shapes, parts, image features, and fixation patterns. Our modeling results offer insight into the nature of the specific mechanisms that are most likely to contribute to image memorability. The brain perceives and remembers using both feedforward and feedback processing, and this processing is modulated by top-down and bottom-up attention. Because of this, it is difficult to attribute an effect like the one we describe to any single mechanism using neural data alone. The fact that variations in response magnitudes that correlate with memorability emerge from static, feed-forward, and fixed networks suggests that memorability variation is unlikely to follow primarily from the types of attentional mechanisms that require top-down processing or plasticity beyond that required for wiring up a system to identify objects.

## Materials and methods

As an overview, three types of data are included in this paper: (1) Behavioral and neural data collected from two rhesus monkeys that were performing a single-exposure visual memory task; (2) Human-based memorability scores for the images used in the monkey experiments, and (3) The responses of units at different layers of three convolutional neural network models trained to classify objects and scenes . The Methods associated with each type of data are described below.

### Behavioral and neural data collected from two rhesus monkeys that were performing a single-exposure visual memory task

Experiments were performed on two naïve adult male rhesus macaque monkeys (*Macaca mulatta*) with implanted head posts and recording chambers. All procedures were performed in accordance with the guidelines of the University of Pennsylvania Institutional Animal Care and Use Committee. Monkey behavioral and neural data were also included in an earlier report that examined the relationship between behavioral reports of familiarity as a function of the time between novel and

familiar presentations (e.g., 'rates of forgetting') and neural responses in IT cortex (*Meyer and Rust, 2018*). The results presented here cannot be inferred from that report.

## The single-exposure visual memory task

All behavioral training and testing were performed using standard operant conditioning (juice reward), head stabilization, and high-accuracy, infrared video eye tracking. Stimuli were presented on an LCD monitor with an 85 Hz refresh rate using customized software (http://mworks-project.org).

Each trial of the monkeys' task involved viewing one image for at least 400 ms and indicating whether it was novel (had never been seen before) or familiar (had been seen exactly once) with an eye movement to one of two response targets. Images were never presented more than twice (once as novel and then as familiar) during the entire training and testing period of the experiment. Trials were initiated by the monkey fixating on a red square (0.25°) on the center of a gray screen, within an invisible square window of ±1.5°, followed by a 200 ms delay before a 4° stimulus appeared. The monkeys had to maintain fixation of the stimulus for 400 ms, at which time the red square turned green (go cue) and the monkey made a saccade to the target indicating that the stimulus was novel or familiar. In monkey 1, response targets appeared at stimulus onset; in monkey 2, response targets appeared at the time of the go cue. In both cases, targets were positioned 8° above or below the stimulus. The association between the target (up vs. down) and the report (novel vs. familiar) was swapped between the two animals. The image remained on the screen until a fixation break was detected. The first image presented in each session was always a novel image. The probability of a trial containing a novel vs. familiar image quickly converged to 50% for each class. Delays between novel and familiar presentations were pseudorandomly selected from a uniform distribution, in powers of two (n-back = 1, 2, 4, 8, 16, 32 and 64 trials corresponding to mean delays of 4.5 s, 9 s, 18 s, 36 s, 1.2 min, 2.4 min, and 4.8 min, respectively).

The images used for both training and testing were collected via an automated procedure that downloaded images from the Internet. Images smaller than 96*96 pixels were not considered and eligible images were cropped to be square and resized to 256*256 pixels. An algorithm removed duplicate images. The image database was randomized to prevent clustering of images according to the order in which they were downloaded. In both the training and testing phases, all images of the dataset were presented sequentially in a random order (i.e. without any consideration of their content). During the testing phase, 'novel' images were those that each monkey had never encountered in the entire history of training and testing. To determine the degree to which these results depended on images with faces and/or body parts, images were scored by two human observers who were asked to determine whether each image contained one or more faces or body parts of any kind (human, animal or character). Conflicts between the observers were resolved by scrutinizing the images. Only 19% of the images used in these experiments contained faces and/or body parts.

The activity of neurons in IT was recorded via a single recording chamber in each monkey. Chamber placement was guided by anatomical magnetic resonance images in both monkeys. The region of IT recorded was located on the ventral surface of the brain, over an area that spanned 5 mm lateral to the anterior middle temporal sulcus and 14–17 mm anterior to the ear canals. Recording sessions began after the monkeys were fully trained on the task and after the depth and extent of IT was mapped within the recording chamber. Combined recording and behavioral training sessions happened 4–5 times per week across a span of 5 weeks (monkey 1) and 4 weeks (monkey 2). Neural activity was recorded with 24-channel U-probes (Plexon, Inc) with linearly arranged recording sites spaced with 100 μm intervals. Continuous, wideband neural signals were amplified, digitized at 40 kHz and stored using the Grapevine Data Acquisition System (Ripple, Inc). Spike sorting was done manually offline (Plexon Offline Sorter). At least one candidate unit was identified on each recording channel, and 2–3 units were occasionally identified on the same channel. Spike sorting was performed blind to any experimental conditions to avoid bias. For quality control, recording sessions were screened based on their neural recording stability across the session, their numbers of visually responsive units, and the numbers of behavioral trials completed. A multi-channel recording session was included in the analysis if: (1) the recording session was stable, quantified as the grand mean firing rate across channels changing less than 2-fold across the session; (2) over 50% of neurons were visually responsive (a loose criterion based on our previous experience in IT), assessed by a visual inspection of rasters; and (3) the number of successfully completed novel/familiar pairs of trials

exceeded 100. In monkey 1, 21 sessions were recorded and six were removed (two from each of the three criteria). In monkey 2, 16 sessions were recorded and four were removed (1, 2 and 1 due to criterion 1, 2 and 3, respectively). The resulting data set included 15 sessions for monkey 1 (n = 403 candidate units), and 12 sessions for monkey 2 (n = 396 candidate units). The sample size (number of successful sessions recorded) was chosen to match our previous work (*Meyer and Rust, 2018*). Both monkeys performed many hundreds of trials during each session (~600–1000, corresponding to ~300–500 images each repeated twice). The data reported here correspond to the subset of images for which the monkeys' behavioral reports were recorded for both novel and familiar presentations (e.g. trials in which the monkeys did not prematurely break fixation during either the novel or the familiar presentation of an image). Finally, units were screened for stimulus-evoked activity via a comparison of their responses in a 200 ms period before stimulus onset ($-200$ ms $-$ 0 ms) versus after stimulus onset (80–280 ms) with a two-sided t-test, p<0.01. This yielded 353 (of 403) units for monkey 1 and 354 (out of 396) units for monkey 2.

Accurate estimate of population response magnitude requires many hundreds of units, and when too few units are included, magnitude estimates are dominated by the stimulus selectivity of the sampled units. To perform our analyses, we thus concatenated units across sessions to create a larger pseudopopulation. In the case of the pooled data, this included 27 sessions in total (15 sessions from monkey 1 and 12 from monkey 2). When creating this pseudopopulation, we aligned data across sessions in a manner that preserved whether the trials were presented as novel or familiar, their n-back separation, and image memorability scores (obtained using Materials and methods described below). More specifically, the responses for each unit always contained sets of novel/familiar pairings of the same images, and pseudopopulation responses across units were always aligned for novel/familiar pairs that contained the same n-back separation and images with similar memorability scores. The number of images that could be included in the pseudopopulation was limited by the session for which the fewest images were obtained.

For the other sessions, a matched number of images were subselected separately for each n-back by ranking images within that n-back by their memorability scores, preserving the lowest-ranked and highest-ranked images within that session, and selecting the number of additional images required as those with memorability scores that were evenly spaced between the two extreme memorability scores for that session. The resulting pseudopopulation consisted of the responses to 107 images presented as both novel and familiar (i.e. 15, 15, 16, 17, 17, 15 and 12 trials at 1, 2, 4, 8, 16, 32 and 64-back, respectively). To perform the neural analyses (*Figure 2b*, *Figure 2—figure supplements 2–3*), a memorability score for each of the 107 pseudopopulation images was computed as the mean of the memorability scores across all the actual images that were aligned to produce that pseudopopulation response. The average standard deviation across the set of memorability scores used to produce each pseudopopulation response was 0.05, where memorability ranges 0–1. To perform behavioral analyses (*Figure 2c*, *Figure 2—figure supplement 4*), the memorability score as well as binary performance values (correct/wrong at reporting that a familiar image was familiar) were retained for each of the 107 images, across each of the 27 sessions. As a control analysis, we created a second pseudopopulation using the same techniques but after excluding the responses to images that contained faces and/or body parts (where the content of each image was determined as described above). The resulting pseudopopulation consisted of the responses to 87 images presented as both novel and familiar. Because only a small fraction of images contained faces or body parts (19%), we were not able to create a pseudopopulation with images that only contained only faces and/or bodies using the same methods applied for the main analysis.

## Human-based memorability scores for the images used in the monkey experiments

We obtained memorability scores for the images used in the monkey experiments using MemNet (*Khosla et al., 2015*) estimates. MemNet is a convolutional neural network (CNN) trained to estimate image memorability on a large-scale dataset of natural images (LaMem; *Khosla et al., 2015*), publicly available at memorability.csail.mit.edu). LaMem consists of 60K images drawn from a diverse range of sources (see *Khosla et al., 2015* for more detail). Each image in this dataset is associated with a memorability score based on human performances in an online memory game on Amazon's Mechanical Turk. Behavioral performances were corrected for the delay interval between first and second presentation to produce a single memorability score for each image. Specifically,

(*Isola et al., 2011*) demonstrated that memorability follows a log-linear relationship as a function of the delay interval between the first and second image presentations, described as:

$$m_t^i = m_T^i + \alpha \log \frac{t}{T}$$

where $m_T^i$ denotes the memorability score for image $i$ after a delay of $T$, $c^i$ represents the base memorability, and $\alpha$ is the memorability decay factor over time. Memorability scores were corrected for delay interval using this equation. After training, MemNet estimates visual memorability of natural images near the upper bound imposed by human performance: MemNet estimates reach 0.64 rank correlation with mean human-estimated memorability, while the upper bound of consistency between human scores has a rank correlation of 0.68. Here we treat MemNet memorability estimates as a proxy for human memorability scores.

The memorability scores were obtained using the network weights reported in *Khosla et al. (2015)* and publicly available at http://memorability.csail.mit.edu/download.html. This network was originally trained using the Caffe framework (*Jia et al., 2014*), and we ported the trained network to Pytorch (*Paszke et al., 2017*) using the caffe-to-torch-to-pytorch package at https://github.com/fanq15/caffe_to_torch_to_pytorch. Before passing images into MemNet, we preprocessed them as described in *Zhou et al. (2014)*: we resized images to 256 × 256 pixels (with bilinear interpolation), subtracted the mean RGB image intensity (computed over the dataset used for pretraining, as described in *Zhou et al., 2014*), and then produced 10 crops of size 227 × 227 pixels. The 10 crops were obtained by cropping the full image at the center and at each of the four corners and by flipping each of these five cropped images about the vertical axis. All 10 crops were passed through MemNet. The average of these 10 scores was used as the mean prediction of the model for the input image. This mean prediction was then linearly transformed to obtain the estimated memorability score:

Memorability_score = min (max ((output - mean_pred)*2 + additive_mean, 0), 1)
where following *Khosla et al. (2015)*, we set mean_pred = 0.7626 and additive_mean = 0.65.

## The responses of units at different layers of CNN models trained to classify objects and scenes

We evaluated the correlation between response magnitude and image memorability on images from the LaMem dataset (*Khosla et al., 2015*) using three commonly used convolutional neural networks (CNNs). All reported models were evaluated on the full test set of split 1 of LaMem, which contains 10,000 images. We chose to use LaMem images, as each image in this dataset is labeled with a memorability score computed directly from human behavioral performance (i.e. not estimated with a model; see above and *Khosla et al., 2015* for details of data collection and memorability score computation). All networks were run in TensorFlow 1.10 (*Abadi et al., 2016*; software available from tensorflow.org), using custom Python evaluation code.

The results presented in *Figure 3* were obtained by running images from this dataset through HybridCNN (*Zhou et al., 2014*). HybridCNN is a network with an identical architecture to AlexNet (*Krizhevsky et al., 2012*). HybridCNN was first trained to classify natural images of objects and scenes using data from the ImageNet Large Scale Visual Recognition Challenge (ILSVRC) 2012, a 1000-way object classification dataset (*Deng et al., 2009*), as well as the Places 183-way scene classification dataset (*Zhou et al., 2014*), for a combined 1183-way classification task. For details of training, see *Zhou et al. (2014)*. Results were obtained using the network weights reported in *Zhou et al. (2014)* and publicly available at http://places.csail.mit.edu/downloadCNN.html. This network was originally trained using the Caffe framework (*Jia et al., 2014*), and we ported the trained network to TensorFlow using the caffe-tensorflow package https://github.com/ethereon/caffe-tensorflow. Random initialization baselines were obtained using the same architecture, but randomly sampling the weights using the initialization algorithm described in *Glorot and Bengio (2010)*.

Before passing images into each network, we preprocessed them as described in *Zhou et al. (2014)* and above: we resized images to 256 × 256 pixels (with bilinear interpolation), subtracted the mean RGB image intensity (computed over the training dataset), and then cropped the central 227 × 227 and passed it into the network. The response magnitude (L2 norm) of each layer was computed over the full output vector of each hidden layer. In all cases, we show the magnitude of hidden layer output after applying the nonlinear operation. Results for the two networks presented

in *Figure 3—figure supplement 1* were obtained in an identical manner, except for the image preprocessing step. For each network, images were preprocessed as described in the original papers (AlexNet: *Krizhevsky et al., 2012*; VGG-16: *Simonyan and Zisserman, 2015*).

For all three networks (HybridCNN, AlexNet, and VGG-16), we computed correlations for all convolutional and fully-connected hidden layers. The Pearson correlation coefficient was used to measure correlation. All correlations were computed over the full set of 10,000 images described above. 95% confidence intervals for the correlation coefficient of each layer were obtained by bootstrapping over the set of 10,000 per-image layer magnitudes and memorability scores. 95% confidence intervals were estimated empirically as the upper and lower 97.5%-centiles of the bootstrapped correlation coefficients for each layer and condition. Bootstrapped resampling was performed independently for each layer and each condition (trained or randomly connected). In all cases, bootstrap estimates were performed using 10,000 samples (with replacement) of the full dataset of 10,000 images. The bootstrapping procedure was also used to conduct one-tailed tests to determine whether the correlations between memorability and response magnitude were stronger in the trained as compared to the randomly initialized network at each layer separately. p-values were estimated by taking pairs of correlation coefficients computed on the bootstrapped data for each condition and measuring the rate at which the correlation for the random layer exceeded the correlation for the trained layer.

## Acknowledgements

This work was supported by the National Eye Institute of the National Institutes of Health (award R01EY020851 to NCR), the Simons Foundation (Simons Collaboration on the Global Brain award 543033 to NCR), and the National Science Foundation (award 1265480 to NCR and award 1532591 to AO).

## Additional information

### Funding

| Funder | Grant reference number | Author |
| --- | --- | --- |
| National Eye Institute | R01EY020851 | Nicole Rust |
| Simons Foundation | 543033 | Nicole Rust |
| National Science Foundation | 1265480 | Nicole Rust |
| National Science Foundation | 1532591 | Aude Oliva |

The funders had no role in study design, data collection and interpretation, or the decision to submit the work for publication.

### Author contributions
Andrew Jaegle, Vahid Mehrpour, Yalda Mohsenzadeh, Conceptualization, Investigation, Writing—original draft, Writing—review and editing; Travis Meyer, Investigation, Writing—review and editing; Aude Oliva, Nicole Rust, Conceptualization, Funding acquisition, Writing—original draft, Project administration, Writing—review and editing

### Author ORCIDs
Andrew Jaegle (iD) https://orcid.org/0000-0003-1698-9901
Vahid Mehrpour (iD) https://orcid.org/0000-0003-1682-5931
Yalda Mohsenzadeh (iD) https://orcid.org/0000-0001-8525-957X
Travis Meyer (iD) http://orcid.org/0000-0003-4672-5368
Aude Oliva (iD) https://orcid.org/0000-0002-6920-914X
Nicole Rust (iD) https://orcid.org/0000-0002-7820-6696

## Ethics

Animal experimentation: All procedures were performed in accordance with the guidelines of the University of Pennsylvania Institutional Animal Care and Use Committee under protocol 804222.

## Decision letter and Author response

Decision letter https://doi.org/10.7554/eLife.47596.014
Author response https://doi.org/10.7554/eLife.47596.015

# Additional files

## Supplementary files

• Transparent reporting form
DOI: https://doi.org/10.7554/eLife.47596.012

## Data availability

Source monkey behavioral and neural data are included in the manuscript as supporting files.

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
