## [Decision Letter]

Thank you for submitting your article "Population response magnitude variation in inferotemporal cortex predicts image memorability" for consideration by *eLife*. Your article has been reviewed by two peer reviewers, and the evaluation has been overseen by a Reviewing Editor and Timothy Behrens as the Senior Editor. The following individuals involved in review of your submission have agreed to reveal their identity: Elizabeth A. Buffalo (Reviewer #2).

The reviewers have discussed the reviews with one another and the Reviewing Editor has drafted this decision to help you prepare a revised submission.

Summary:

The reviewers were impressed with several aspects of the work, including linking the strength of the overall population response in IT to memorability and the idea that coding schemes for memorability and identity may be orthogonal. However, they raised a number of issues that should be addressed, before the paper will be considered for publication in *eLife*. These are listed below.

Essential revisions:

1) Please provide the rationale for not utilizing the monkey behavioral data to directly link the strength and variability of responses to behavior. The question how the monkey IT was trained in classification needs to be clarified.

2) Please address the question of the role of learning and experience in making some images more memorable. A related question is whether IT population response depends on whether the stimuli are novel or familiar. This important question cannot be addressed with responses that are averaged across the two groups of stimuli, as it is currently the case.

3) Please clarify the approach used to create pseudopopulations since some of the data was likely to be collected for simultaneously recorded neurons and combined with the data collected on other days.

4) Please address the question whether model predictions can be used to modify images to increase their memorability.

5) Please discuss whether the face and body parts are the features with the highest positive correlation to memorability.

6) Please address the effect of categorization training on correlations between the model responses on image memorability and on the population response magnitude.

*Reviewer #1:*

In this paper, Jaegle and colleagues explore the connection between population responses in IT cortex and image memorability. Their hypothesis is that images that evoke a stronger population response will correlate positively with the likelihood of being able to remember that image. Using a combination of model predictions prom human experiments and their IT recordings, they obtain evidence supporting this hypothesis. Taken together, the authors suggest that prior attempts to reduce variability in population responses may have missed this potentially important connection.

The basic message in this paper is relatively clear, with the notion that overall response levels might both be variable and that this variability can be directly linked to behavior is intriguing. The paper is weakened considerably by the fact that the monkey behavioral data are not really used to draw these conclusions (Figure 2C does present a version of this, but it doesn't really form the basis for the core analyses). Indeed, the authors have a nicely designed behavioral paradigm, and we don't even see the effect of delay on the monkeys' performance. The main correlations are between the memorability predictions from the model designed to predict behavioral in human participants and evoked pseudopopulation response magnitudes. While the inability to clearly connect the behavioral data to the neural data in the same experiment is somewhat disappointing, the authors do point out that the relation may be explained by the development of representations that are more generally a results of having to classify objects, which could be preserved between species. This may be true, but exactly how the monkey's IT cortex has been trained in classification is not entirely clear.

A related question that the paper raises that is not directly addressed is how specific learning or training could alter their findings. It would seem that specialization of neural representations resulting from training could make certain images more memorable than others as a function of experiences. While it seems like the authors are assuming that they are sampling a large space where no subset of image would likely be more familiar or experienced than others, it might be worth at least discussing how experience might affect their results. Further, the authors clearly didn't target any particular cell populations, but they could have landed in a cluster of specialized cells (faces or other categories). It's not clear how their hypotheses would account for this?

Another point that I thought could have been briefly discussed is the relation of this work to the large literature of studies of memory formation, wherein activity at the time of encoding can affect subsequent recall. This work is clearly distinct from that, but that distinction could be made more clear.

I found the description of the creation of the pseudopopulations a bit difficult to follow (Materials and methods section), which leaves me somewhat concerned about the conclusions that are drawn from these. What happens when the sessions with 20 or so sites are used to make predictions, as these populations are real and simultaneously collected under identical conditions?

Do the authors have any approach for using their network predictions to help modify images to make them more or less memorable? This would be a powerful manipulation if it directly affected neural responses.

*Reviewer #2:*

In this manuscript, Jaegle et al. examine the hypothesis that variation in the magnitude of the population response of neurons in the monkey inferotemporal (IT) cortex is positively correlated with image memorability. The authors analyze a pseudo population of 707 IT units recorded as monkeys performed a recognition memory task with a large set of complex images, with each image presented once as novel and once as familiar. The data suggests that while image identity can be decoded through the population vector direction (as shown in DiCarlo et al., 2012), image memorability correlates with the response magnitude of the population, with higher magnitude population response reflecting greater image memorability. In addition, the authors probed CNN models trained to categorize objects and found that larger magnitude responses in higher layers of the network correlated with images that had high memorability. The question of how neural activity in IT contributes to recognition memory is not well understood, and the hypothesis that memorability and identity may be related to orthogonal population responses is novel and intriguing. The manuscript is well-written and should be of interest to a wide audience. I have just a few suggestions to improve clarity.

1) In constructing the population response, each neuron's response to a given image is averaged across the novel and familiar presentation. Because memorability is defined here as a bottom-up feature of the stimuli, it would be interesting to know whether there was a change in the IT population response for novel vs familiar presentations. If experience affected the response, then, presumably, the response to just the novel presentation would provide the strongest correlation with memorability.

2) The data in Khosla et al. suggests that memorability is strongly related to salience, in that stimuli with more consistent fixation locations across subjects had higher memorability. It would be helpful to discuss the extent to which memorability is thought to reflect something beyond image salience. That is, would it be expected that IT population response magnitude (and the monkey's recognition memory performance) would also be positively correlated with a measure of image salience?

3) Khosla et al. also suggest that CNN features with the highest positive correlation to memorability correspond to faces and body parts. It would be helpful to discuss whether this relationship to IT stimulus selectivity may underlie the relationship between IT population response magnitude and the memorability index.

4) The authors report a correlation between responses in higher layers of the CNN and image memorability after categorization training. It would be interesting to know what effect the categorization training had on population responses, i.e., did the training produce a separation in population magnitude across images that was more consistent early in training or was there a more complex relationship between training and population response magnitude?

---

## [Author Response]

Essential revisions:1) Please provide the rationale for not utilizing the monkey behavioral data to directly link the strength and variability of responses to behavior. The question how the monkey IT was trained in classification needs to be clarified.

We have clarified in the Results, “In contrast to the human-based scores, which reflect the estimated average performance of ~80 human individuals, our monkey behavioral data are binary (i.e. correct/wrong for each image). As such, the monkey behavioral data cannot be used in the same way to concatenate neural data across sessions to create a pseudopopulation sufficiently large to accurately estimate IT population response magnitudes.” We also note that because of the species difference, our main claims, which report a correlation between monkey IT neural responses and human behavior, should reasonably reflect a lower bound on the correlation between monkey IT neural responses and monkey behavior.

As for the question of how the monkeys were trained, we have clarified in the Materials and methods that these experiments were performed in two naïve monkeys that were not previously trained to perform some other laboratory task. Additionally, we now elaborate that the images used for these experiments were downloaded via an automated procedure from the Internet, and after randomizing the image sequence in the database, monkeys were trained and tested by marching through the database without regard to image content. Finally, we have stated in the Results, “While the monkeys involved in these experiments were not explicitly trained to report object identity, they presumably acquired the ability to identify objects naturally over their lifetimes.”

(Received as an Update to Essential Revision #1 after we received the other reviews): Further to this, one of the reviewers had an additional comment regarding point #1 that the editors have asked us to pass on to you: "I had one question about point #1 – my reading is that the monkeys were trained on the novel/repeat task and the CNN was trained on object classification. Is this question related to the monkeys or the CNN?"

We are a bit confused by what “this question” is referring to here. To paraphrase our best understanding of the question at hand, “Monkeys were trained to perform a visual familiarity task whereas the CNN was not trained to report visual familiarity, rather, the CNN was trained to perform object classification. Do your results inform us about the neural representations of visual familiarity or the neural representations of object classes?” To ensure that there is no confusion behind the rationale supporting our CNN analysis, we have rephrased that paragraph in the Results. In sum, the CNN results suggest that a system trained to categorize images produces response magnitude variation in IT-analogous network layers that correlate with image memorability, even though this type of CNN cannot remember anything (i.e. after training and the network parameters are frozen, it responds the same way to an image despite how many times it has ‘seen’ it). In the discussion, we interpret these results in a fairly straightforward way, tying into the literature that documents that when the visual representation of an image is more robust, performance on visual memory tasks is better. However, we emphasize that our report of IT response magnitude variation with the class of images is novel and non-trivial, and our results demonstrate that this variation has behavioral consequences (on image memorability).

2) Please address the question of the role of learning and experience in making some images more memorable. A related question is whether IT population response depends on whether the stimuli are novel or familiar. This important question cannot be addressed with responses that are averaged across the two groups of stimuli, as it is currently the case.

We now report the correlations for novel and familiar images separately (r = 0.62 vs. r = 0.58, respectively), which were both strong, highly significant, and very similar to one another. Regarding understanding the role that experience plays in memorability, very little is known about it. To quote a non-author colleague, “The topic of experience as it relates to memorability is still unexplored, but is a high priority for future memorability research.” (Bainbridge, 2019; Psychology of Learning and Motivation).

3) Please clarify the approach used to create pseudopopulations since some of the data was likely to be collected for simultaneously recorded neurons and combined with the data collected on other days.

We have clarified both why we create a pseudopopulation, “Accurate estimate of population response magnitude requires many hundreds of units, and when too few units are included, magnitude estimates are dominated by the stimulus selectivity of the sampled units.” as well as how the pseudopopulation was created (Materials and methods).

4) Please address the question whether model predictions can be used to modify images to increase their memorability.

Such a model has just been introduced (by one of the senior authors of our paper, Aude Oliva, and her colleagues) and we now refer to it, “Future work will be required to explore the ultimate bounds of image memorability variation, possibly via the use of newly developed generative adversarial models that create images with increased or decreased memorability (Goetschalckx et al., 2019).”

5) Please discuss whether the face and body parts are the features with the highest positive correlation to memorability.

To address whether clusters of specialized cells, including faces and body parts, could account for our results, we now include the results of a control analysis in which images with that content were eliminated. The correlation between memorability scores and IT response magnitude remained strong when images containing faces and/or bodies were excluded from the analysis (r=0.62) suggesting that our results are not an artifactual consequence of recording from patches of neurons enriched for face or body selectivity. We have also explained in the Materials and methods, “Because only a small fraction of images contained faces or body parts (19%), we were not able to create a pseudopopulation with images that only contained only faces and/or bodies using the same methods applied for the main analysis.”

6) Please address the effect of categorization training on correlations between the model responses on image memorability and on the population response magnitude.

Regarding the role of learning and experience: we have analyzed the trajectories of correlation strength between response magnitude and image memorability as a function of training for one of the neural networks, described in detail below. These results demonstrate that the correlations generally increase and remain high with training, consistent with our main findings.

Reviewer #1:In this paper, Jaegle and colleagues explore the connection between population responses in IT cortex and image memorability. Their hypothesis is that images that evoke a stronger population response will correlate positively with the likelihood of being able to remember that image. Using a combination of model predictions prom human experiments and their IT recordings, they obtain evidence supporting this hypothesis. Taken together, the authors suggest that prior attempts to reduce variability in population responses may have missed this potentially important connection.The basic message in this paper is relatively clear, with the notion that overall response levels might both be variable and that this variability can be directly linked to behavior is intriguing. The paper is weakened considerably by the fact that the monkey behavioral data are not really used to draw these conclusions (Figure 2C does present a version of this, but it doesn't really form the basis for the core analyses). Indeed, the authors have a nicely designed behavioral paradigm, and we don't even see the effect of delay on the monkeys' performance.

We agree that whenever conditions allow, it is always better to correlate behavioral and neural data within the same species. At the same time, it is worth noting the limitations that follow from the single-trial nature of this particular question (i.e. single-exposure visual memory). We now clarify in the Results “In contrast to the human-based scores, which reflect the estimated average performance of ~80 human individuals, our monkey behavioral data are binary (i.e. correct/wrong for each image). As such, the monkey behavioral data cannot be used in the same way to concatenate neural data across sessions to create a pseudopopulation sufficiently large to accurately estimate IT population response magnitudes.” Measuring monkey memorability scores would be fantastic, but it would require training and testing many more monkeys than we have reasonable access to (e.g. at least 40 monkey individuals). Similarly, recording from many hundreds of units simultaneously in monkey IT (such that aligning data across sessions to create a pseudopopulation is not required) would be fantastic, but that slightly beyond the technology available today. In sum, our data present a reasonable way to address the hypothesis presented in Figure 1 in the context of current limitations in the field.

Additionally, to re-emphasize a point presented above (Essential revisions point 1): because of the species difference, our main claims, which report a correlation between monkey IT neural responses and human behavior, should reasonably reflect a lower bound on the correlation between monkey IT neural responses and monkey behavior. Additionally, as the reviewer notes, we present evidence for a correlation between human and monkey behavior in Figure 2C.

The main correlations are between the memorability predictions from the model designed to predict behavioral in human participants and evoked pseudopopulation response magnitudes. While the inability to clearly connect the behavioral data to the neural data in the same experiment is somewhat disappointing, the authors do point out that the relation may be explained by the development of representations that are more generally a results of having to classify objects, which could be preserved between species. This may be true, but exactly how the monkey's IT cortex has been trained in classification is not entirely clear.

We have clarified in the Materials and methods that these experiments were performed in two naïve monkeys that were not previously trained to perform some other laboratory task. We have clarified, “While the monkeys involved in these experiments were not explicitly trained to report object identity, they presumably acquired the ability to identify objects naturally.” We have also elaborated the Materials and methods to explain that the images used for these experiments were downloaded via an automated procedure from the Internet and after randomizing the image sequence in the database, monkeys were trained and tested by marching through the database without regard to image content.

We have also addressed the issue in the Discussion, where we state, “The correlation between what humans and monkeys find memorable (Figure 2C) is at first pass surprising in light of the presumed differences in what typical humans and monkeys experience. However, understanding that memorability variation emerges in CNNs trained for object categorization (Figure 3), coupled with the similarities in object representations between humans and monkeys (Rajalingham et al., 2015), provides insight into the preservation of memorability correlations across these two primate species.”

A related question that the paper raises that is not directly addressed is how specific learning or training could alter their findings. It would seem that specialization of neural representations resulting from training could make certain images more memorable than others as a function of experiences. While it seems like the authors are assuming that they are sampling a large space where no subset of image would likely be more familiar or experienced than others, it might be worth at least discussing how experience might affect their results.

Regarding understanding the role that experience plays in memorability, so incredibly little is known about it that we can’t say anything beyond the most obvious speculation. To quote a nonauthor colleague, “The topic of experience as it relates to memorability is still unexplored, but is a high priority for future memorability research.” (Bainbridge, 2019; Psychology of Learning and Motivation).

Further, the authors clearly didn't target any particular cell populations, but they could have landed in a cluster of specialized cells (faces or other categories). It's not clear how their hypotheses would account for this?

This is an important point. To address whether clusters of specialized cells, including faces and body parts, could account for our results, we included the results of a control analysis in which images with faces and body parts not included. The correlation between memorability scores and IT response magnitude remained strong when images containing faces and/or bodies were excluded from the analysis (Pearson correlation: r=0.62) suggesting that our results are not an artifactual consequence of recording from patches of neurons enriched for face or body selectivity, thereby producing higher responses to this (highly memorable) image class.

Another point that I thought could have been briefly discussed is the relation of this work to the large literature of studies of memory formation, wherein activity at the time of encoding can affect subsequent recall. This work is clearly distinct from that, but that distinction could be made more clear.

We now state in the Discussion, “Our work relates to ‘subsequent memory effects’ whereby indicators of higher neural activity in structures both within and outside the medial temporal lobe during memory encoding are predictive of more robust remembering later on (reviewed by Paller and Wagner, 2002). To tease apart whether the origin of memorability could be attributed to optimizations for visual as opposed to mnemonic processing, we […]”

I found the description of the creation of the pseudopopulations a bit difficult to follow (Materials and methods section), which leaves me somewhat concerned about the conclusions that are drawn from these. What happens when the sessions with 20 or so sites are used to make predictions, as these populations are real and simultaneously collected under identical conditions?

We have revised the description of the creation of the pseudopopulation in the Materials and methods to make it more clear. In terms of analyzing the simultaneously recorded data, we have explored this question. When analyzing data recorded from a single session, variability in IT population response magnitude is dominated by the selectivity of those units for image identity (e.g. the fact that a particular subset of units prefers some images over others). It is only when the responses of large numbers of units are included that this sampling bias averages out and the correlations with image memorability are observed. We have clarified in both the Results and Materials and methods that the accurate estimate of population response magnitude requires many hundreds of units.

Do the authors have any approach for using their network predictions to help modify images to make them more or less memorable? This would be a powerful manipulation if it directly affected neural responses.

Agreed! We now include the following sentence addressing this approach in the Discussion:

“Future work will be required to explore the ultimate bounds of image memorability variation, possibly via the use of newly developed generative adversarial models that create images with increased or decreased memorability (Goetschalckx et al., 2019).” To be clear, the technology now exists, but these model predictions remain untested.

Reviewer #2:In this manuscript, Jaegle et al. examine the hypothesis that variation in the magnitude of the population response of neurons in the monkey inferotemporal (IT) cortex is positively correlated with image memorability. The authors analyze a pseudo population of 707 IT units recorded as monkeys performed a recognition memory task with a large set of complex images, with each image presented once as novel and once as familiar. The data suggests that while image identity can be decoded through the population vector direction (as shown in DiCarlo et al., 2012), image memorability correlates with the response magnitude of the population, with higher magnitude population response reflecting greater image memorability. In addition, the authors probed CNN models trained to categorize objects and found that larger magnitude responses in higher layers of the network correlated with images that had high memorability. The question of how neural activity in IT contributes to recognition memory is not well understood, and the hypothesis that memorability and identity may be related to orthogonal population responses is novel and intriguing. The manuscript is well-written and should be of interest to a wide audience. I have just a few suggestions to improve clarity.1) In constructing the population response, each neuron's response to a given image is averaged across the novel and familiar presentation. Because memorability is defined here as a bottom-up feature of the stimuli, it would be interesting to know whether there was a change in the IT population response for novel vs familiar presentations. If experience affected the response, then, presumably, the response to just the novel presentation would provide the strongest correlation with memorability.

We have now included, “[…] at the same time that IT neural responses exhibited repetition suppression for familiar as compared to novel image presentations (mean proportional reduction in this spike count window = 6.2%; see also (Meyer and Rust, 2018)), the correlation remained strong when computed for the images both when they were novel (Pearson correlation: r = 0.62; p = 2x10-12) as well as when they were familiar (Pearson correlation: r = 0.58; p = 8x10-11).” Our interpretation of these results is that the correlation is strong and highly significant for both novel and familiar images. We note that while there is in fact a small reduction in the correlation for familiar (r=0.58) as compared to novel (r=0.62) images, at least some component of this reduction is likely to follow from the fact that familiar images evoked lower responses, as a consequence of repetition suppression. While determining the functional form of repetition suppression is tricky with our single-trial data, others have described repetition suppression as having a multiplicative impact on IT neural responses, and multiplicative rescaling would produce the result we observe here (i.e. similar correlations for novel and familiar responses).

2) The data in Khosla et al. suggests that memorability is strongly related to salience, in that stimuli with more consistent fixation locations across subjects had higher memorability. It would be helpful to discuss the extent to which memorability is thought to reflect something beyond image salience. That is, would it be expected that IT population response magnitude (and the monkey's recognition memory performance) would also be positively correlated with a measure of image salience?

We have included a discussion of this in the Discussion. We conclude that salience and memorability are partially overlapping but not one and the same.

3) Khosla et al. also suggest that CNN features with the highest positive correlation to memorability correspond to faces and body parts. It would be helpful to discuss whether this relationship to IT stimulus selectivity may underlie the relationship between IT population response magnitude and the memorability index.

This is an important point. To address whether clusters of specialized cells, including faces and body parts, could account for our results, we included the results of a control analysis in which images with faces and body parts not included. The correlation between memorability scores and IT response magnitude remained strong when images containing faces and/or bodies were excluded from the analysis (r=0.62) suggesting that our results are not an artifactual consequence of recording from patches of neurons enriched for face or body selectivity, thereby producing higher responses to this (highly memorable) image class.

4) The authors report a correlation between responses in higher layers of the CNN and image memorability after categorization training. It would be interesting to know what effect the categorization training had on population responses, i.e., did the training produce a separation in population magnitude across images that was more consistent early in training or was there a more complex relationship between training and population response magnitude?

We have retrained one of the networks, AlexNet, from a randomly initialized state to perform the requested analysis (Author response image 1). These results reveal that the average correlation (across layers) between response magnitude and memorability rises and then saturates, and that this saturation happens at an earlier point training than the saturation of object categorization performance (Author response image 1, left panel). This is true for all layers with the exception of Conv5, which peaks early and declines slightly with training, whereas the layers both above and below it all saturate and remain high (Author response image 1, right panel). In sum, the training dynamics reveal a general trend of increases in correlation with training followed by saturation, and these results are thus consistent with our main findings. Having confirmed that, we have opted not to include this plot in the manuscript because the analysis touches on a host of questions that lie beyond the scope of our work. Namely, while our work complements existing reports of representational similarities between visual brain areas and fully trained convolutional neural networks (CNNs), we do not wish to imply (or even speculate) that the developmental trajectories that brains and CNNs take to get to their fully trained states are also similar, particularly in light of the biologically implausible learning rules used to train CNNs (e.g. reviewed by Bengio et al., ‘Towards Biological Plausible Deep Learning, 2016, arXiv). Rather, the developmental trajectory of the neural correlates of image memorability is an important topic for future work.

**Author response image 1. respfig1:** Emergence of the correlation between response magnitude and memorability in one network, AlexNet, as a function of training duration starting from a randomly initialized network. *Left:* Shown for the left-hand axis (black) is the mean Pearson correlation (across all network layers) as a function of training, where training is quantified by the numbers of images (in Millions) used for training and 0 reflects the randomly initialized state. Superimposed with the right-hand axis (blue) is the object categorization performance of the network, quantified as top-1 and top-5 accuracy on a 1000-way object categorization task on the ImageNet validation set (see below for more details). *Right:* Shown are the Pearson correlations for the individual layers of the network that were combined to produce the mean in the right panel (black).

Details of experiment shown in Author response image 1: To examine the effect of object classification training on the correlation between population activation norm and image memorability, we trained AlexNet from scratch on the ImageNet training set and evaluated the correlation in the network periodically through training, using checkpoints collected once an hour. We trained AlexNet using the AlexNet-v2 implementation in TensorFlow-Slim (cited below). We trained the network using stochastic gradient descent on a 1000-way cross-entropy classification loss using a batch size of 32, an initial learning rate of 0.01, and L2 weight decay with a weight of 4e-5. We decayed the learning rate every two epochs through training by a multiplicative factor of 0.94. We trained the network until the classification loss on the ImageNet validation set stopped decreasing.

We evaluated correlation with memorability by evaluating networks on the LaMem test set (as for all network experiments in the paper). To examine the effect of object classification training on the network memorability correlation, we additionally evaluated the classification accuracy of the network during training, using the same checkpoints used to measure the memorability correlation. We measured classification accuracy using the top-1 and top-5 classification accuracy rates, both of which are standard for evaluating performance on ImageNet. The top-1 accuracy is computed as the fraction of images on the validation set on which the network outputs the largest probability for the correct class label. The top-5 accuracy is computed as the fraction of images for which the network predicts the true class as one of the five highest probability classes. Both of these measures can be computed directly from network output because the network is trained for 1000-way classification (which is standard for DNN training on ImageNet) and outputs class probabilities for all 1000 classes for each evaluated image. We report both measures on the ImageNet validation set, which is not used for training.

"TensorFlow-Slim: a lightweight library for defining, training and evaluating complex models in TensorFlow" S. Guadarrama, N. Silberman, 2016. https://github.com/tensorflow/tensorflow/tree/master/tensorflow/contrib/slim